# The Search for an Interesting Partner to Combine with PD-L1 Blockade in Mesothelioma: Focus on TIM-3 and LAG-3

**DOI:** 10.3390/cancers13020282

**Published:** 2021-01-14

**Authors:** Elly Marcq, Jonas R. M. Van Audenaerde, Jorrit De Waele, Céline Merlin, Patrick Pauwels, Jan P. van Meerbeeck, Scott A. Fisher, Evelien L. J. Smits

**Affiliations:** 1Center for Oncological Research (CORE), Integrated Personalized and Precision Oncology Network (IPPON), University of Antwerp, 2610 Wilrijk, Belgium; jonas.vanaudenaerde@uantwerpen.be (J.R.M.V.A.); jorrit.dewaele@uantwerpen.be (J.D.W.); celine.merlin@uantwerpen.be (C.M.); patrick.pauwels@uza.be (P.P.); jan.vanmeerbeeck@uza.be (J.P.v.M.); evelien.smits@uza.be (E.L.J.S.); 2Department of Pathology, Antwerp University Hospital, 2650 Edegem, Belgium; 3Department of Pulmonology & Thoracic Oncology, Antwerp University Hospital, 2650 Edegem, Belgium; 4National Centre for Asbestos Related Diseases (NCARD), Perth 6008, Australia; scott.fisher@uwa.edu.au; 5School of Biomedical Sciences, University of Western Australia, Perth 6008, Australia

**Keywords:** immune checkpoints, mesothelioma, in vitro, in vivo, TIM-3, LAG-3, PD-1, PD-L1

## Abstract

**Simple Summary:**

Malignant pleural mesothelioma is an aggressive cancer most commonly associated with asbestos exposure. Its prognosis is very poor, and the current treatments have only a limited impact on survival. Therefore, there is an urgent need to develop new treatment strategies. We investigated combinations of different immune checkpoint blocking antibodies that have already shown promising results in several cancer types. We suggested that the effect of a combination treatment might even be better than one, single therapy. Based on our in vitro results about the secretion of several immune related molecules we selected promising treatments for further investigation in a mesothelioma mouse model. We found that monotherapy with an immune checkpoint blocking antibody against programmed death-1 (PD-1), and its combination with another blocking antibody against lymphocyte activation gene-3 (LAG-3), resulted in delayed tumor growth and survival benefit in our mice. Further research is warranted to optimize the treatment schedule of the combination therapy.

**Abstract:**

Malignant pleural mesothelioma (MPM) is an aggressive cancer that is causally associated with previous asbestos exposure in most afflicted patients. The prognosis of patients remains dismal, with a median overall survival of only 9–12 months, due to the limited effectiveness of any conventional anti-cancer treatment. New therapeutic strategies are needed to complement the limited armamentarium against MPM. We decided to focus on the combination of different immune checkpoint (IC) blocking antibodies (Abs). Programmed death-1 (PD-1), programmed death ligand-1 (PD-L1), T-cell immunoglobulin mucin-3 (TIM-3), and lymphocyte activation gene-3 (LAG-3) blocking Abs were tested as monotherapies, and as part of a combination strategy with a second IC inhibitor. We investigated their effect in vitro by examining the changes in the immune-related cytokine secretion profile of supernatant collected from treated allogeneic MPM-peripheral blood mononuclear cell (PBMC) co-cultures. Based on our in vitro results of cytokine secretion, and flow cytometry data that showed a significant upregulation of PD-L1 on PBMC after co-culture, we chose to further investigate the combinations of anti PD-L1 + anti TIM-3 versus anti PD-L1 + anti LAG-3 therapies in vivo in the AB1-HA BALB/cJ mesothelioma mouse model. PD-L1 monotherapy, as well as its combination with LAG-3 blockade, resulted in in-vivo delayed tumor growth and significant survival benefit.

## 1. Introduction

Immune checkpoints (ICs) are immune modulatory molecules responsible for maintaining immune homeostasis, modulating immune responses and preventing autoimmunity [1]. ICs recently gained lot of interest in the field of cancer immunotherapy because their overexpression reduces antitumor immune responses resulting in an immune resistant tumor [1]. Therefore, blocking ICs in order to abolish tumor immune resistance became a promising approach for cancer immunotherapy. 

Cytotoxic T-lymphocyte antigen 4 (CTLA-4) was the first IC to be clinically targeted. Promising results [2] stimulated the identification of new ICs as possible targets for cancer immunotherapy. In the first place, we will focus on the IC, programmed death-1 (PD-1), and its ligand, programmed death ligand-1 (PD-L1). In addition, two “less known” ICs, lymphocyte activation gene-3 (LAG-3) and T-cell immunoglobulin mucin-3 (TIM-3), will be investigated.

To date, combinations have been described with chemotherapy and radiotherapy in addition to surgery for the treatment of malignant pleural mesothelioma (MPM) [3]. The series of side effects of these currently used treatments, and their modest impact on median survival, emphasize the need for novel treatment strategies. The rising potential of immunotherapy, due to the crucial role of ICs in tumor immune tolerance, might provide new insights into the treatment of MPM. Several studies have already tried to combine these targeted therapies with other therapies in different types of cancer, and some of them have provided promising results. For example, a significant improvement of progression-free survival (and higher overall response rates were observed in patients with non-small cell lung carcinoma (NSCLC) treated with anti PD-1 (pembrolizumab) plus chemotherapy [4,5]. 

Few studies have investigated combined immune checkpoint blockade (ICB) protocols in MPM. The fact that tumor cells can inhibit antitumor immune responses via different mechanisms suggests that a more effective antitumor immune response might be elicited by blocking multiple ICs. The major question that remains is which combination might be the best. Combining ipilimumab with nivolumab demonstrated higher response rates compared to ipilimumab monotherapy in advanced melanoma and NSCLC [6,7,8]. However, treatment-related adverse events (AEs) (high grade) were more frequent in the combination group. The INITIATE trial and MAPS-2 trial also showed the efficacy of ipilimumab/nivolumab in MPM patients, with a tolerable safety profile [9,10]. Combined blockade with PD-1 and LAG-3 or TIM-3 has also gained interest due to the preclinical reported enhanced antitumor immune responses compared to monotherapy [11,12]. The combination of antibodies (Abs) blocking PD-1 and LAG-3 is also of great interest due to their expression on tumor infiltrating lymphocytes (TILs), thereby restricting T-cell responses to the tumor microenvironment (TME), and which might lead to a favorable safety profile. Based on the encouraging results and promising prospects of combined blockade we decided to investigate the combination of IC blocking Abs in MPM.

## 2. Results

### 2.1. Cytokine Secretion Is Treatment-Dependent

Activated T-cells secrete cytokines in order to activate other immune cells, and to exert their cytotoxic function towards tumor cells. MPM cells of the epithelioid (NCI-H2818, NCI-H2795) and sarcomatoid (NCI-H2731) subtypes were placed in co-cultures with peripheral blood mononuclear cells (PBMC) from healthy donors. To determine the T-cell activity in these co-cultures after 72 h of treatment with IC blocking Abs, supernatant of the co-cultures was collected to look at the secretion of seven immune-related cytokines and one immune-related enzyme, using ELISA and multiplex electrochemiluminescence. Only the cytokines and enzyme with significant differences between the treatment conditions are shown for each of our three MPM cell lines (Figure 1, Figure 2, Figure 3, Figure 4 and Figure 5). 

ELISA was performed for IFNγ and granzyme B. IFNγ is secreted by activated T-cells and natural killer (NK) cells, providing information about the effect of our different treatments on T- and NK-cell activation. Granzyme B, secreted by activated cytotoxic T-cells and natural killer cells, induces apoptosis in target cells. The highest concentrations of IFNγ were found after treatment with anti PD-1 or anti PD-L1, and their combination with anti TIM-3 (Figure 1). Although significant differences were only found for the NCI-H2818 cell line, similar trends for IFNγ secretion were observed for the other two cell lines. Regarding granzyme B secretion, the highest concentrations were found for the same treatments as for IFNγ (Figure 2).

Multiplex electrochemiluminescence was performed to investigate the secretion of IL-2, IL-5, IL-6, IL-10, IL-1*β,* and TNF-*α*. Significant differences were found for IL-2, IL-5, and IL-10 (see Figure 3, Figure 4 and Figure 5).

IL-2 is secreted by activated T-cells, and stimulates the differentiation of regulatory T-cells, effector T-cells, and memory T-cells. It regulates immune responses through a negative and positive feedback loop. Highest concentrations were detected for the anti PD-1 and anti PD-L1 monotherapies, and their combination with TIM-3 blockade (Figure 3), similarly to our findings for IFNγ and granzyme B. 

IL-5 plays an important role in the stimulation of B-cell growth and eosinophil activation. The highest concentration of IL-5 was observed with the combined treatment of anti PD-1 and anti TIM-3 for two of our cell lines (NCI-H2795 and NCI-H2731), and for the combination of anti PD-L1 with anti LAG-3 for the NCI-H2818 cell line (Figure 4).

IL-10 enhances the proliferation and survival of B-cells. It can also act as an anti-inflammatory molecule by downregulating the expression of co-stimulatory molecules on immune cells. As noted for IFNγ, granzyme B, and IL-2, the highest concentrations of IL-10 were observed in the supernatant of co-cultures treated with anti PD-1 and anti PD-L1 monotherapies, and their combination with TIM-3 blockade (Figure 5).

IL-6, IL-1*β*, and TNF-*α* were also found in all our supernatant samples. We did not observe clear differences for IL-6 or IL-1*β* concentrations between the different treatments. In contrast, as described for IFNγ, granzyme B, IL-2, and IL-10, the highest concentrations of TNF-*α* (a pro-inflammatory cytokine) were observed in the supernatant of co-cultures treated with anti PD-1 and anti PD-L1 monotherapies, and their combination with TIM-3 blockade, although there were no significant differences found for our three MPM cell lines.

In summary, our results showed that PD-1 blockade or PD-L1 blockade, and their combination with a TIM-3 blocking antibody, induced the highest concentration of IFNγ, granzyme B, IL-2, IL-10, and TNF-*α*. Although the differences between the treatment conditions were not all statistically significant, clear trends could be observed. These results suggested anti TIM-3 with anti PD-1 or anti PD-L1 as interesting treatment strategies for further in vivo investigation.

### 2.2. Co-Culture with MPM Cells Influences the Immune Checkpoint Expression Profile of PBMC

In order to rationally select a combination strategy for validation of our in vitro data, and for further in vivo research, we investigated if tumor cells can influence the IC expression profile of PBMC. Flow cytometry was used to analyze the expression of PD-1, PD-L1, TIM-3, and LAG-3 on PBMC of different healthy donors before, and 72 h after, co-culture with MPM cells. Similar results were found for co-cultures with our three different MPM cell lines. While there was nearly no PD-1 expression on PBMC before and after co-culture, the expression of its ligand PD-L1 was significantly upregulated after co-culture with two of our MPM cell lines (NCI-H2818 and NCI-H2795). For the third cell line the same trend was observed. We also saw that TIM-3 expression was slightly decreased after co-culture, while the expression of LAG-3 remained practically the same (Figure 6).

Considering the significant upregulation of PD-L1 on healthy donor PBMC after co-culture with MPM tumor cells, we chose the combination of anti PD-L1 with anti TIM-3 for further in vivo investigation in the AB1-HA BALB/cJ mesothelioma mouse model. 

### 2.3. In Vivo Results

To validate our choice for the combination of anti PD-L1 with anti TIM-3 in vivo, we looked at their expression on AB1-HA tumor cells before in vivo injection. These data confirmed that PD-L1 and TIM-3, were expressed on AB1-HA cells before injection, suggesting that blocking these proteins might be a combination strategy with therapeutic potential in vivo (Figure 7).

Mice were treated with anti PD-L1, anti TIM-3, or anti LAG-3 as monotherapies or with the combination of anti PD-L1 + anti TIM-3, or anti PD-L1 + anti LAG-3. The latter was included as a “control” for the validation of our in vitro results because the lowest levels of immune-related cytokines and granzyme B were detected for anti-LAG-3 in combination with anti PD-L1. 

All mice were treated with three doses of blocking antibodies given every third day (q3dx3). One dose consisted of 200 μg IC blocking antibody, either given as a monotherapy or in combination with a second IC blocking antibody. 

We observed a delay in tumor growth for anti PD-L1 and anti PD-L1 + anti LAG-3 treatment when compared to our PBS control group, though not statistically significant (Figure 8A). When looking at the Kaplan–Meier curve, a statistically significant survival benefit was observed for those same treatments in comparison to the PBS control (Figure 8B) (PD-L1 vs. PBS, *p* = 0.03; PD-L1 + LAG-3 vs. PBS, *p* = 0.003). 

Interestingly, our best in vivo treatments (anti PD-L1, anti PD-L1 + anti LAG-3) only partially corresponded to our in vitro data (anti PD-L1 and anti PD-L1 + anti TIM-3). We hypothesized that the discrepancy between our in vitro and in vivo results might be due to differences in the IC expression profile of lymphocytes in vitro versus in vivo. Therefore, we harvested tumors and spleens from BALB/cJ mice before the first treatment and we looked at the expression of PD-1, PD-L1, TIM-3, and LAG-3 on CD4 and CD8 effector lymphocytes in the spleen as well as on tumor infiltrating lymphocytes (TILs). 

Based on the differences in mean fluorescence intensity (ΔMFI), we observed higher expression for all ICs on tumor resident CD4+ and CD8+ TILs relative to the spleen (Figure 9A). When looking at the percentages of positive cells we clearly saw that more CD8+ effector T cells that express PD-L1 and LAG-3 were found in the tumor compared to the spleen. The same observation was made for LAG-3 expression on CD4+ T helper cells in the tumor (Figure 9B). Though a higher intensity of PD-1 expression was found on TILs, only a low percentage of cells were positive. More TILs expressed TIM-3 at a higher intensity compared to T cells in the spleen, though the percentages of TIM3+ TILs were remarkably lower than PD-L1+ and LAG-3+ TILs (Figure 9B). 

## 3. Discussion

New therapeutic strategies are needed to complement the present armamentarium against MPM, because the current first line chemotherapy only has a limited impact on overall survival in MPM [13,14,15]. Overexpression of ICs reduces antitumor immune responses and is suspected to promote an immune resistant tumor [1]. Therefore, we believe that abolishing this tumor immune resistance by blocking several IC pathways is a promising approach for the development of new MPM treatments, and that is why we focused on combined IC blockade (ICB). The baseline expression of immune checkpoints and their ligands on MPM cells and PBMC in co-culture was investigated in our lab [16]. Those data confirmed us that the direct targets (being PD-1, PD-L1, TIM-3, and LAG-3) of our treatments were expressed in co-cultures, and this therefore supported continuing our research on novel combination strategies that included those targets.

As described in the literature, ICs are immunomodulatory molecules responsible for maintaining immune homeostasis and modulating immune responses through a series of inhibitory signaling pathways [1]. Overexpression of ICs has already been described in several cancer types, and results in suppression of immune cell activity [17]. ICB aims at enhancing antitumor immune responses by preventing immunosuppression. As extensively reported by Shi et al. [18], IC expression can be regulated through the secretion of several cytokines, such as IFNγ, IL-5, and IL-10. Since antitumor responses are regulated by a complex interplay between pro-inflammatory and anti-inflammatory cytokines, we examined whether there were differences in the cytokine secretion profile and granzyme B in the supernatant of MPM–PBMC co-cultures treated with different IC blocking Abs. We found that PD-1 and PD-L1 monotherapies, as well as their combination with TIM-3 blockade, were associated with the highest secretion of immune related-cytokines and granzyme B. Increased gene expression or cytokine secretion levels for IFNg, TNFa, IL-10, and granzyme B in response to IC therapy have also been observed in vivo in AB1-HA and AE17 mesothelioma mouse models [19], as well as in other murine tumor models, such as B16 melanoma and MC38 colon carcinoma [20,21]. Although IFNγ, IL-2, IL-10, and TNF-α are described as pro-inflammatory [22,23], it should be kept in mind that IL-2 and IL-10 function via a negative feedback loop, and therefore they can also have an anti-inflammatory function depending on the context [24,25]. This might explain the difference between our in vitro and in vivo findings, discussed below. 

Interestingly, our in vivo results did not completely correspond to our in vitro findings. Although the targets of our “most promising” in vitro treatments, PD-L1 and TIM-3 blockade, were expressed on the AB1-HA cells, only the PD-L1 monotherapy showed a significant survival benefit in vivo compared to the PBS control (*p* = 0.03). Surprisingly, although LAG-3 expression was absent on our murine MPM cells, and the expression on human PBMC remained steady before and after co-culture, the combination with PD-L1 also showed a significant survival benefit in vivo (*p* = 0.003). This can be explained by the high percentage of PD-L1+ and LAG-3+ TILs that we observed after harvesting the tumor and spleen. Remarkably, while only very low percentages of PD-L1+ and LAG-3+ CD8+ effector T cells were found in the spleen, very high percentages of PD-L1+ and LAG-3+ TILs were found. These data support the idea that the TME influences the IC expression profile of immune cells, which might influence the response to immunotherapy, as already described in the literature [18]. AB1-HA cells are MHCII negative, as reported by Marzo et al. [26], though this does not explain the differences between our in vitro and in vivo results. When blocking LAG-3 one would expect to enhance the functioning of MHCII+ APCs, resulting in a better anti-tumor immune response. Differentiation and activation of APCs might mainly occur at the draining lymph node, and this is a process that cannot be mimicked in the in vitro experimental set-up that we used. This might explain the difference between our in vitro and in vivo findings.

We observed a lower mean fluorescence intensity (MFI) value, but a higher overtone % for PD-1 on T cells in the spleen compared to the TILs (Figure 9). This means that PD-1 expression on T cells in the spleen was less intense, but PD-1 was expressed on more cells in the spleen compared to TILs. In contrast, Kwiecien et al. [27] looked at PD-1 expression on T cells in bronchoalveolar lavage fluids (BALF) and blood samples (PB) of NSCLC patients, and observed a significantly higher proportion of CD8+ and CD4+ cells with PD-1 expression in the BALF when compared with the PB. Ahmadzadeh et al. [28] examined PD-1 expression on TILs in metastatic melanoma lesions. They found that both the frequency and the level of expression of PD-1 were significantly higher on CD4 and CD8 TILs compared with T cells in normal tissues and peripheral blood in the same patients and healthy donors. Our results also showed a higher MFI value for PD-1 expression on TILS, and therefore are consistent with the observation of Ahmadzadeh et al. [28]. The fact that we did not observe a higher frequency of PD-1+ TILs might be due to less immune cell infiltration in mesothelioma compared to melanoma and NSCLC. One should also keep in mind that our expression data are just a snapshot, since IC expression is known to be very dynamic.

A significant survival benefit in comparison to our PBS control was observed for PD-L1 monotherapy and its combination with anti LAG-3 (*p* = 0.03; *p* = 0.003). Both treatments did not differ significantly from each other, suggesting that the survival benefit of the combination strategy was mainly due to PD-L1 blockade. However, the combination of PD-L1 and LAG-3 blockade differed significantly more from the PBS control than the PD-L1 monotherapy, suggesting that LAG-3 has some effect on the PD-L1 monotherapy, though not an additive one. Survival data, alone, are not sufficient to conclude whether PD-L1 + LAG-3 blockade is more effective in comparison to PD-L1 monotherapy. Further research is warranted with a larger sample size. The examination of differences in the immune infiltrate in mice treated with monotherapy compared to the combination strategy might provide us with more information about the possible added value of LAG-3 blockade to PD-L1 blockade.

We are the first to report in vivo data on the combination of anti PD-L1 with TIM-3 or LAG-3 blockade in mesothelioma. Our findings for combined PD-L1 and LAG-3 blockade are supported by Everett et al. [29] who reported a positive influence on tumor kinetics for this combination strategy in a colon carcinoma mouse model. The fact that TIM-3 blockade does not show as promising results on in vivo tumor kinetics as expected might be due to the fact that TIM-3 possibly has a more activating, rather than an inhibitory, effect, as reported in the literature [30,31]. This means that TIM-3 blockade results in less immune activation, which is consistent with our data showing an increased tumor growth and worse survival compared to our control.

One should keep in mind that our in vitro and in vivo experiments had some limitations. For the in vitro co-cultures an allogeneic setting was used due to the limited availability of patient-matched PBMC and tumor cells. We also used a subcutaneous tumor model for our in vivo experiments instead of an orthotopic model, because this allowed a more convenient monitoring of the tumor kinetics over time. Overall, we tried to mimic the “in patient” situation as well as possible, though one cannot neglect the impact of the tumor microenvironment (TME) on the treatment outcome, which is confirmed by our results.

Assessing IC (ligand) expression on PBMC is only one way to assess potential IC candidates. A therapy checkerboard approach, as described by Fear et al. [32], might be another option. Moreover, looking at the IC expression on TILs from tumor bearing mice might also be a possibility, but would be unlikely to alter our observed in vivo results. It is known that PD-1 blockade and TIM-3 blockade do nothing as monotherapies or in combination in AB1-HA bearing mice [32], though they are great at revitalizing exhausted T cells in viral models and relatively effective in some clinical settings. In the end there is no best way to identify IC combinations, ultimately it has to be done empirically.

## 4. Materials and Methods 

### 4.1. Cell Lines

Three immortalized human MPM cell lines, kindly provided to us by the Netherlands Cancer Institute (NKI), were used to investigate the effectiveness of several therapies in vitro. The two most important histological subtypes of MPM are represented by these cell lines. The NCI-H2818 and NCI-H2795 cell lines belong to the epithelioid subtype, which is the most common, while the NCI-H2731 represents the sarcomatoid, most lethal subtype. Cell lines were cultured in flasks with DMEM/F-12 Glutamax^TM^ medium supplemented with 10% fetal bovine serum (FBS). Cells were harvested with 0.05% trypsin. 

The murine AB1-HA cell line was used for our in vivo research [26,33]. Dr. Scott Fisher from the Tumor Immunology Group at the National Centre for Asbestos related diseases (NCARD) at the University of Western Australia in Perth kindly provided it to us. AB1-HA cells were maintained in RPMI 1640 medium (Gibco©, Life Technologies, Belgium, Ghent) supplemented with 10% FBS and 50 mg/mL Geneticin (G418, Life Technologies, Belgium, Ghent). Cells were harvested with 0.05% trypsine, washed two times in phosphate buffered saline (PBS), and counted before in vivo inoculation. Only cell suspensions with a viability of >95%, as assessed by trypan blue exclusion assay, were used.

### 4.2. Peripheral Blood Mononuclear Cells (PBMC)

PBMC were isolated out of blood from healthy donors, which is obtained from the blood transfusion center in Mechelen, Antwerp (ethical committee approval 14/39/397). Isolation of these PBMC was done using a Ficoll density gradient centrifugation [16]. After isolation the PBMC were used for allogeneic co-cultures with MPM cell lines

### 4.3. Immune Checkpoint Blocking Antibodies—In Vitro

For our in vitro experiments the PD-1 (nivolumab) and PD-L1 (durvalumab) blocking Abs were kindly provided by BMS and AstraZeneca, respectively. For TIM-3 and LAG-3 research grade blocking Abs from Thermo Fisher Scientific were used. Concentrations of the blocking Abs (10 μg/mL) were based on those described in the literature.

### 4.4. Flow Cytometry

The effect of the different (combined) immunotherapies on the IC expression profile of healthy donor human PBMC was investigated in vitro before and after co-culture with our three different MPM cell lines using flow cytometry. Expression of PD-1, PD-L1, TIM-3, and LAG-3 was checked on PBMC before co-culture, and 72 h later after co-culture and treatment in a 6-well plate at an effector:target (E:T) ratio of 20:1. Abs for our targets and their corresponding isotypes for flow cytometry were all conjugated with PE and bought from BD Biosciences^®^ (Belgium, Erembodegem): PD-1 (CD279, clone MIH4), PD-L1 (CD274, clone MIH1), TIM-3 (CD366, clone 7D3), and LAG-3 (CD223, clone T47-530). The expression of PD-1, PD-L1, TIM-3, and LAG-3 was determined on the murine AB1-HA cells prior to in vivo injection, and on single cell suspensions from in vivo tumors and spleen before the start of treatment (tumor size ± 25 mm^2^). Single cell suspensions were prepared as described by Van Audenaerde et al. [34]. Single stainings for PD-1 (clone 29F.1A12), PD-L1 (clone 10F.9G2), TIM-3 (clone RMT3-23), and LAG-3 (clone C9B7W) expression on the AB1-HA cell line were done using anti mouse Abs from Biolegend^®^ (The Netherlands, Amsterdam) conjugated with PE. A multicolor panel was used for the staining of our in vivo tumors and spleens. The following anti mouse Abs were included: CD45.1-APC Cy7 (clone A20), CD8-FITC (clone 53-6.7), CD4-PercP-Cy5 (clone RM4-5), CD3-PE (clone 17A2), PD-1-BV42 (clone 29F.1A12), PD-L1-APC (clone 10F.9G2), TIM-3-BV785 (clone RMT3-23), and LAG-3-PeCy7 (clone C9B7W). The CD4 and CD8 Abs were bought from Thermo Fisher Scientific^®^ (Belgium, Ghent), all the other Abs were from Biolegend^®^ (The Netherlands, Amsterdam).

### 4.5. Allogeneic Co-Cultures 

#### For ELISA and Electrochemiluminescence

Activity of T-cells within the PBMC was investigated by looking at the secretion of several cytokines in the supernatant of co-cultures. Three different MPM cell lines were used (NCI-H2818, NCI-H2795, and NCI-H2731). MPM cells were plated, this time in a 6-well plate, and IC blocking Abs were added at least one hour prior to adding PBMC. PBMC were placed in co-culture with tumor cells at an E:T ratio of 20:1. As a positive control for our cytokine secretion, a well with only PBMC was pre-stimulated with 2.5 µg/mL PMA + 0.5 µg/mL ionomycin 48 h after being in co-culture with MPM cells and 24 h before the final analysis. Seventy-two hours later, the 6-well plate was centrifuged at 453 g for 5 min after which the supernatant was collected and stored at −20 °C for further analysis.

### 4.6. Enzyme-Linked Immunosorbent Assay (ELISA)

To determine the interferon-gamma (IFN-γ) and granzyme B concentration in the collected supernatant, a human IFN-γ ELISA kit from Thermo Fisher Scientific and a human granzyme B DuoSet ELISA kit from R&D Systems were used [35]. Next, 50 µL of supernatant (consisting of cell-secreted proteins) was transferred in duplo to microtiter plates that were pre-coated overnight with IFN-γ or granzyme B capture Abs. Analyses were performed according to the manufacturers’ protocol. OD-values of each well were measured with an IMARK^®^ (Bio-rad, Belgium, Temse) microplate absorbance plate reader (λ = 450 nm). In order to determine the IFN-γ and granzyme B concentrations in our supernatant, standards (with known IFN-γ or granzyme B concentrations) were included. 

### 4.7. Electrochemiluminescence

A multiplex electrochemiluminescent assay (U-PLEX 6-assay, Meso Scale Discovery (MSD©, Belgium, Brussels)) was used for the detection of interleukin (IL)-1 beta, IL-2, IL-5, IL-6, IL-10, and tumor necrosis factor-alpha (TNF-α). Analyses were performed according to the manufacturers’ protocol (MSD©). In summary, 25 μL of supernatant was added to a multiplex plate to detect all of the previously mentioned cytokines at once. Biotinylated capture Abs were coupled to cytokine specific linkers that bind to unique spots on the plate. Similarly to the ELISA, the sample is then loaded, after which electrochemiluminescent-labeled detection Abs were added. Finally, the plates were loaded in an MSD instrument that applied a voltage to electrodes at the bottom of the plate causing the labeled capture Abs to emit light. Quantification of the cytokines is based in the intensity of the emitted light, which is proportional to the amount of cytokine present in the sample. Data analysis was done using MSD Discovery workbench 4.0 software.

### 4.8. Mice

Female BALB/cJ mice (6–8 weeks) were bought from Janvier and maintained under standard housing conditions at the animal facility of the University of Antwerp. The experiments were conducted in accordance with the University of Antwerp animal ethics guidelines and protocols (animal wellbeing ethical committee approval ECD 2014-78) and with the European code for the care and use of animals for scientific purposes.

### 4.9. Tumor Cell Inoculation

Mice were inoculated subcutaneously (s.c.) into the shaved flank with 2 × 10^5^ AB1-HA cells. Calipers were used to monitor tumor size. By multiplying width and length the tumor size was determined in square millimeters (mm^2^). Once the tumor size exceeded 150 mm^2^ mice were euthanized in accordance with animal ethics guidelines.

### 4.10. Immune Checkpoint Blocking Antibodies—In Vivo

For our in vivo experiments all Abs were bought from BioXcell: anti CTLA-4 (clone 9H10), anti PD-L1 (clone 10F.92G), anti TIM-3 (clone RMT3-23), and anti LAG-3 (clone C9B7W). The concentrations of the blocking Abs and the treatment schedule (100 μg or 200 μg per dose, 1 dose every 3 days, 3 doses in total) were based on the literature [32] and personal communication with prof. Dr. Scott Fisher from the National Center for Asbestos-Related diseases (NCARD) in Perth, Australia. The IC blocking Abs were added simultaneously when used in a combination strategy.

Based on data published by Fear et al. [32] we included anti CTLA-4 as a “positive” control for our in vivo experiments, as shown in Appendix A. This data confirmed that our AB1-HA data were reproducible and that our mouse model was responsive to ICB.

### 4.11. Preparation of Single Cell Suspensions

Tumors and spleens were harvested prior to treatment in order to prepare single cell suspensions for flow cytometric analysis. Single cell suspensions from the spleens were obtained by mechanical digestion with the back of a 5 mL syringe, after which they were passed through a 40 µm cell strainer (BD Biosciences^®^, Belgium, Erembodegem) and lysis buffer was added. Cells were centrifuged at 453 g for 5 min at 4 °C and resuspended in PBS. Again, the suspension was passed through a 40 µm cell strainer (BD Biosciences^®^, Belgium Erembodegem) and centrifuged at 453 g for 5 min at 4 °C, resulting in single cell suspensions of the spleen.

Tumors were minced into small tissue fragments of 1–2 mm², suspended into tumor digestion medium (RPMI 1640 + 10% FBS + 0.2 mg/mL DNA-se I + 0.5 mg/mL liberase + 2.5 mg/mL collagenase D) and incubated for 30 min at 37 °C and 5% CO_2_ on a cell rocker. After incubation, tissue pieces were further mechanically disrupted by fierce pipetting and pressed trough a 70 µL cell strainer. Tumor cell suspensions were washed with PBS to remove the remaining digestion medium, after which the tumor single cell suspensions were ready for further downstream applications.

### 4.12. Statistics

Significant differences in IC expression profile before and after co-culture were investigated with a paired-sample t-test. One-way ANOVA was used to check if there were significant differences in cytokine secretion between the untreated control sample and the different treatments. The non-parametric Kruskal–Wallis test and a Tukey post-hoc test was used to investigate differences between the treatments in vitro. *p*-values ≤ 0.05 were considered significant. GraphPad Prism was used for the analysis.

Kaplan–Meier curves were made and a log-rank test was performed to detect differences in survival between the different treatment groups in vivo. A one-way ANOVA with Tukey post-hoc analysis in SPSS software was used to look at differences in tumor kinetics.

## 5. Conclusions

Our in vitro results show that PD-1 blockade or PD-L1 blockade and their combination with a TIM-3 blocking antibody resulted in the highest concentration of cytokines and granzyme B. Further in vivo investigation in the AB1-HA BALB/c mesothelioma mouse model showed the best responses for PD-L1 monotherapy and its combination with anti LAG-3. Interestingly, antitumor responses are regulated by a complex interplay between pro-inflammatory and anti-inflammatory cytokines, of which the secretion can be influenced by the interaction between tumor cells and immune cells present in the TME. In turn, cytokines can regulate the IC expression in the TME. Our in vivo data highlight the importance of the TME in the light of treatment response. We found some promising results, though there is still room for improvement. By doubling the doses, we obtained better results. Our next step will be to adjust the dosing schedule in order to see whether this can further enhance treatment response.

## Figures and Tables

**Figure 1 cancers-13-00282-f001:**
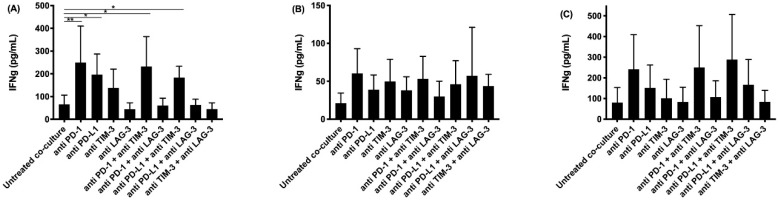
IFNγ secretion in supernatant collected after co-culturing peripheral blood mononuclear cells (PBMC) with malignant pleural mesothelioma (MPM) cells and treatment with immune checkpoint (IC) blocking antibodies (Abs). The amount of IFNγ present in supernatant collected after 72 h was determined using an ELISA assay. Supernatant was collected from co-cultures of the MPM cell lines (**A**) NCI-H2818, (**B**) NCI-H2795, and (**C**) NCI-H2731 with human PBMC. The experiments were repeated three times with three different PBMC donors (*n* = 3). Results were compared to our negative and positive controls. Error bars represent the standard deviation. * *p* < 0.05, ** *p* < 0.01: significant difference in IC expression.

**Figure 2 cancers-13-00282-f002:**
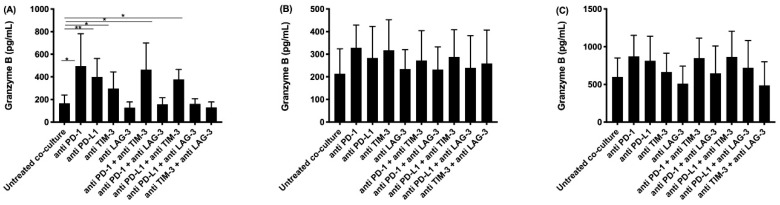
Granzyme B secretion in supernatant collected after co-culturing PBMC with MPM cells and treatment with IC blocking Abs. The amount of granzyme B present in supernatant collected after 72 h was determined using an ELISA assay. Supernatant was collected from co-cultures of the MPM cell lines (**A**) NCI-H2818, (**B**) NCI-H2795, and (**C**) NCI-H2731 with human PBMC. The experiments were repeated three times with three different PBMC donors (*n* = 3). Results were compared to our negative and positive controls. Error bars represent the standard deviation. * *p* < 0.05, ** *p* < 0.01: significant difference in IC expression.

**Figure 3 cancers-13-00282-f003:**
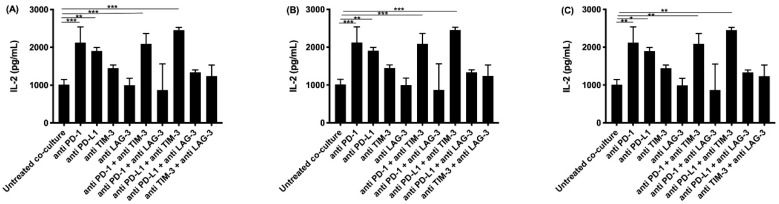
IL-2 secretion in supernatant collected after co-culturing PBMC with MPM cells and treatment with IC blocking Abs. The amount of IL-2 present in supernatant collected after 72 h was determined using multiplex electrochemiluminescence. Supernatant was collected from co-cultures of the MPM cell lines (**A**) NCI-H2818, (**B**) NCI-H2795, and (**C**) NCI-H2731 with human PBMC. The experiments were repeated three times with three different PBMC donors (*n* = 3). Results were compared to our negative and positive controls. Error bars represent the standard deviation. * *p* < 0.05, ** *p* < 0.01, *** *p* < 0.001: significant difference in IC expression.

**Figure 4 cancers-13-00282-f004:**
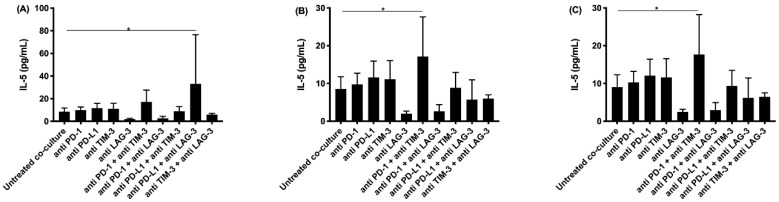
IL-5 secretion in supernatant collected after co-culturing PBMC with MPM cells and treatment with IC blocking Abs. The amount of IL-5 present in supernatant collected after 72 h was determined using multiplex electrochemiluminescence. Supernatant was collected from co-cultures of the MPM cell lines (**A**) NCI-H2818, (**B**) NCI-H2795, and (**C**) NCI-H2731 with human PBMC. The experiments were repeated three times with three different PBMC donors (*n* = 3). Results were compared to our negative and positive controls. Error bars represent the standard deviation. * *p* < 0.05: significant difference in IC expression.

**Figure 5 cancers-13-00282-f005:**
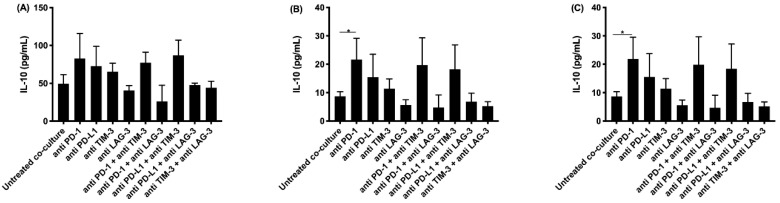
IL-10 secretion in supernatant collected after co-culturing PBMC with MPM cells and treatment with IC blocking Abs. The amount of IL-10 present in supernatant collected after 72 h was determined using multiplex electrochemiluminescence. Supernatant was collected from co-cultures of the MPM cell lines (**A**) NCI-H2818, (**B**) NCI-H2795, and (**C**) NCI-H2731 with human PBMC. The experiments were repeated three times with three different PBMC donors (*n* = 3). Results were compared to our negative and positive controls. Error bars represent the standard deviation. * *p* < 0.05: significant difference in ICP expression.

**Figure 6 cancers-13-00282-f006:**
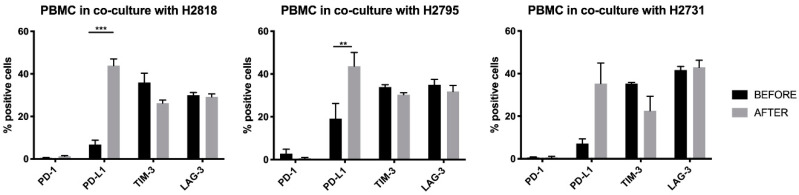
IC expression profile on PBMC before and after allogeneic co-cultures with MPM tumor cells. PBMC were placed in co-culture with tumor cells in a six well plate at an E:T ratio of 20:1. Flow cytometry was used to look at the expression of programmed death-1 (PD-1), programmed death ligand-1 (PD-L1), T-cell immunoglobulin mucin-3 (TIM-3), and lymphocyte activation gene-3 (LAG-3) before and 72 h after allogeneic co-cultures. The experiments were repeated three times with three different PBMC donors (*n* = 3). Error bars represent the standard deviation. ** *p* < 0.01, *** *p* < 0.001: significant difference in IC expression.

**Figure 7 cancers-13-00282-f007:**
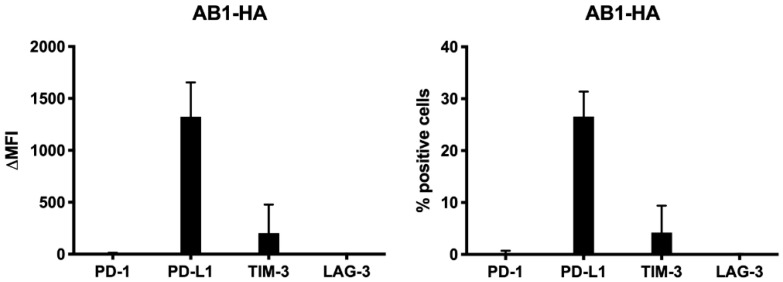
IC expression on the murine AB1-HA cell line. Flow cytometry was used to look at the expression of PD-1, PD-L1, TIM-3, and LAG-3 before subcutaneous (s.c.) injection in vivo. Analyses were repeated three times at different time points. Error bars represent the standard deviation.

**Figure 8 cancers-13-00282-f008:**
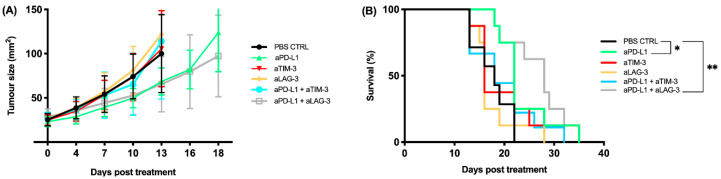
Tumor kinetics and survival data from mice treated with 200 μg of IC blocking antibody. Mice were treated every third day for a total of three doses (q3dx3). (**A**) The tumor growth curves show a decreased tumor growth for anti PD-L1 and anti PD-L1 + anti LAG-3. (**B**) The Kaplan–Meier curve shows a significant survival benefit for anti PD-L1 and anti PD-L1 + anti LAG-3. Error bars represent the standard deviation, *n* = 18, pooled data from three independent experiments, * *p* < 0.05, ** *p* < 0.01: statistically significant difference in survival.

**Figure 9 cancers-13-00282-f009:**
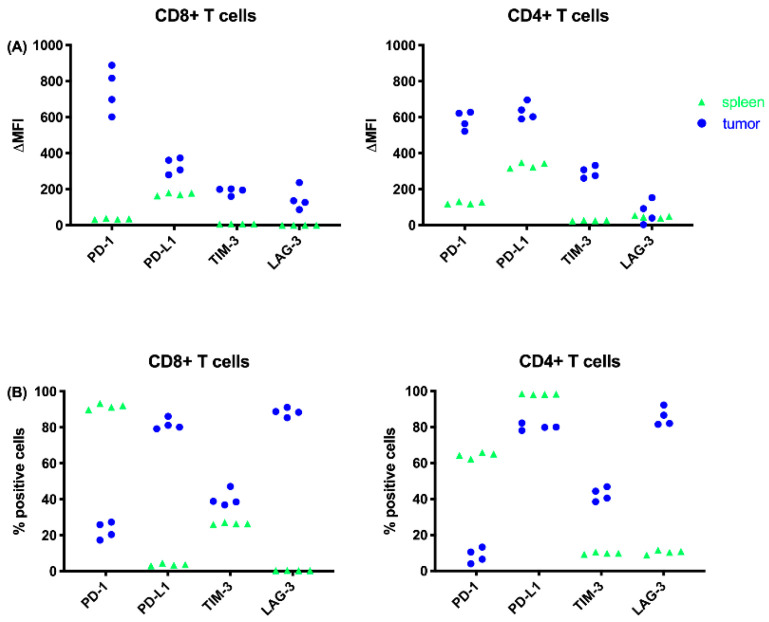
IC expression on T cells depicted as (**A**) delta mean fluorescence intensity (ΔMFI), or (**B**) % positive cells from spleens and tumors harvested from the AB1-HA MPM mouse model prior to treatment. BALB/cJ mice were inoculated with 2 × 10^5^ AB1-HA cells. When tumor size reached ± 25 mm^2^ the mice were sacrificed and tumors and spleens were harvested for flow cytometric analysis (*n* ≥ 8).

## Data Availability

Raw data were generated at the Center for Oncological Research at the University of Antwerp, Belgium. The data that support the findings of this study are available on request from the corresponding author.

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
