# Peer review of "The Search for an Interesting Partner to Combine with PD-L1 Blockade in Mesothelioma: Focus on TIM-3 and LAG-3"

_cancers, 2021, doi:10.3390/cancers13020282_

Round 1

Reviewer 1 Report

The authors used an allogeneic co-culture screen between allogeneic PBMCs and mesothelioma tumor lines in the presence or absence of anti-checkpoint inhibitors (CPI) antibodies in vitro to select antibody combinations. Antibodies selected in vitro were not predictive of their in vivo activity. The idea of selecting anti-CPI antibodies in vitro is a good one, but there is a lack of control for interpreting in vitro experiments and this may explain the failure of the in vivo experiment. Different experiments should be performed to improve the quality of this manuscript

- As the anti-PD-1/anti-CTLA-4 combination has shown some efficacy in mesothelioma (Scherpereel A Lancet Oncol 2019)(Wright K Oncology 2020), in order to validate their experimental condition and show the interest of new combinations, it would be important to evaluate this combination in vitro and in vivo.

- In order to interpret the in vitro experiments, it would be rational to better characterize the tumor lines used for the expression of CPI ligands (PD-L2, HLA class II, galectin 9...). Before and after co-culture It is just mentioned in figure 6 that PD-1, PD-L1, Tim-3, Lag3 are expressed and that the AB1-HA line expresses PD-L1 and Tim-3. Does AB1-HA express Lag3 ligands to explain the in vivo results ?.

- How the authors explain that the addition of anti-Lag3 to anti-PD-1 inhibits in vitro proliferation. Does Lag3 by binding to HLA-Class II molecules inhibit the allogeneic reaction.

- It is indicated that in the majority of cases the combination of anti-PD-1 and anti-Tim-3 is synergistic in vitro. In fact this is not often the case outside of IL-5, which may explain the in vivo results.

- In Figure 9, it is surprising that PD-1 expression on T cells is higher in the spleen than in the tumor. This contradicts what is generally observed in the literature (Kwiecien I et al cancers 2019)(Ahmadzadeh M Blood 2009).

- In Figure 8, the number of mice used and how many times the experiment was performed is not indicated.

Reviewer 2 Report

In this paper, the authors reported the effects in vitro by examining the altered immune-related cytokine secretion profile of supernatant collected from treated allogeneic MPM-PBMC co-cultures and anti-tumor effect in vivo using the combination of different immune checkpoint (IC) blocking antibodies (Abs) such as against PD-1, PD-L1, TIM-3 and LAG-3, on malignant mesothelioma (MM).

The IC blocking Abs have been available for patients with MM. However the effectiveness of IC blocking Abs may be varied in MM patients. These molecular mechanisms of varied effects of IC blocking Abs are unknown, so some data in this paper may be important in immune-oncology. However there are some defects in data and descriptions. Criticisms regarding this revised paper are discussed below.

Comments

  1. Allogeneic assay in vitro

These allogeneic experiments contain PBMC derived from healthy donor and three MM cell lines in this paper. The detail and expressions of HLA class II in APCs healthy donor and these cell lines should be described because APC ligand for LAG3 is HLA class II molecule. Are there different in pattern of cytokine productions by NCI-H2818, NCI-H2795 or NCI-H2731  cells by human PBMC with varied HLA haplotypes?

  1. in vivo assay using AB1-HA cells

The in vivo results using AB1-HA cells are shown in Fig 7 and Fig 8. However there were no data of in vitro/in vivo cytokine productions by treatment with IC blocking Abs. Please reveal some data of in vitro/in vivo cytokine secretions by T cells similar to Fig 1 - 6.

  1. T cell / APC in AB1-HA tumor

The authors reveal the proportions of IC molecules expression of CD4/CD8 T cells in spleen vs tumor. However there is no description of T cell and its subtype and APC in tumor. Please show immunological state in tumor of AB1-HA cells such as infiltration of T cell and its subtype, APC or NK cells.

Round 2

Reviewer 1 Report

The authors responded to my various criticisms. I just find it a pity not to include the figure with the anti-CTLA-4 experiment because in order to evaluate the interest of new combinations it is important to show the results obtained with the reference antibody.

Reviewer 2 Report

In this paper, the authors reported the effects in vitro by examining the altered immune-related cytokine secretion profile of supernatant collected from treated allogeneic MPM-PBMC co-cultures and anti-tumor effect in vivo using the combination of different immune checkpoint (IC) blocking antibodies (Abs) such as against PD-1, PD-L1, TIM-3 and LAG-3, on malignant mesothelioma (MM).

The IC blocking Abs have been available for patients with MM. The effectiveness of IC blocking Abs may be varied in MM patients. These molecular mechanisms of varied effects of IC blocking Abs are unknown, so some data in this paper may be important in immune-oncology. However there are some defects in data and descriptions in revised article. Criticisms regarding this revised paper are discussed below.

Comments

  1. Allogeneic assay in vitro

1) There is a different in pattern of cytokine productions by NCI-H2818, NCI-H2795 or NCI-H2731 cells, especially different secretions of IFNg, granzyme B, IL-5 and IL-10 co-cultured with NCI-H2818 cells. On the other hand, the pattern of IC positives on human PBMCs seem to be similar in co-culture of PBMCs with NCI-H2818, NCI-H2795 or NCI-H2731 cells (Fig 6). Why ?

2) The induction of PD-L1 expression on PBMCs was obvious in three MPM cell lines after co-culture with MPM cells (Fig 6). On the other hand, there are varied cytokine secretions by anti-PD-L1 (Fig 1-5). Why is there discrepancy between Fig 1-5 and Fig 6 ?

  1. in vivo assay using AB1-HA cells

The in vivo results using AB1-HA cells are shown in Fig 7 and Fig 8. However there were no data of in vitro/in vivo cytokine productions by treatment with IC blocking Abs. The authors reply in vitro cytokine productions are shown in human system on Fig 1-5, however data in Fig 1-5 is co-culture system using human MPM and human PBMCs. In vivo data is using murine AB1-HA cells and murine immune cells. Please reveal some data of in vitro/in vivo cytokine secretions by murine T cells similar to Fig 1-6 or depict a summary of refered papers such as Fear V et al., Oncoimmunology, 2018; Zemek R et al., Sci Transl MEd, 2019.

Round 3

Reviewer 2 Report

In this paper, the authors reported the effects in vitro by examining the altered immune-related cytokine secretion profile of supernatant collected from treated allogeneic MPM-PBMC co-cultures and anti-tumor effect in vivo using the combination of different immune checkpoint (IC) blocking antibodies (Abs) such as against PD-1, PD-L1, TIM-3 and LAG-3, on malignant mesothelioma (MM). The IC blocking Abs have been available for patients with MM. However the effectiveness of IC blocking Abs may be varied in MM patients. These molecular mechanisms of varied effects of IC blocking Abs are unknown, so data in this paper may be important in immune-oncology. The authors answered the questions pointed out and revealed the reply to three comments
such as 1)allogeneic assay in vitro, 2) in vivo assay using AB1-HA cells, 3) T cell / APC in AB1-HA tumor in this revised manuscript.